# Impact of secretin receptor homo-dimerization on natural ligand binding

Kaleeckal G. Harikumar[1], Sarah J. Piper [2,3], Arthur Christopoulos [2,3], Denise Wootten [2,3] ✉, Patrick M. Sexton [2,3] ✉ & Laurence J. Miller [1] ✉

Class B G protein-coupled receptors can form dimeric complexes important for high potency biological effects. Here, we apply pharmacological, biochemical, and biophysical techniques to cells and membranes expressing the prototypic secretin receptor (SecR) to gain insights into secretin binding to homo-dimeric and monomeric SecR. Spatial proximity between peptide and receptor residues, probed by disulfide bond formation, demonstrates that the secretin N-terminus moves from adjacent to extracellular loop 3 (ECL3) at wild type SecR toward ECL2 in non-dimerizing mutants. Analysis of fluorescent secretin analogs demonstrates stable engagement of the secretin C-terminal region within the receptor extracellular domain (ECD) for both dimeric and monomeric receptors, while the mid-region exhibits lower mobility while docked at the monomer. Moreover, decoupling of G protein interaction reduces mobility of the peptide mid-region at wild type receptor to levels similar to the mutant, whereas it has no further impact on the monomer. These data support a model of peptide engagement whereby the ability of SecR to dimerize promotes higher conformational dynamics of the peptide-bound receptor ECD and ECLs that likely facilitates more efficient G protein recruitment and activation, consistent with the higher observed functional potency of secretin at wild type SecR relative to the monomeric mutant receptor.

The propensity, stability, and functional importance of dimerization of G protein-coupled receptors (GPCRs) appears to differ between major subfamilies, with class C GPCRs having the highest propensity to form structurally specific and functionally important stable complexes, class A GPCRs able to only transiently form such complexes with variable functional importance, and class B intermediate between these families[1]. The most stable and most extensively studied class B GPCR dimeric complexes involve the secretin receptor (SecR), both as homo-dimers and hetero-dimers with other members of this family[2,3]. This dimerization is structurally symmetrical along the lipid face of transmembrane segment 4 (TM4), contributing to high potency

biological activity at the receptor homo-dimer[4]. This is a feature shared by multiple class B GPCRs. Mutations along the lipid face of TM4 of SecR can disrupt homo-dimeric complexes of that receptor[5], with analogous mutations made in GLP-1R[6] and the calcitonin receptor[7] shown to similarly disrupt their dimerization.

The functional importance of this phenomenon has been demonstrated not only in engineered model cell systems[5], including low levels of receptor expression[8,9], but also in native expression settings[2]. Hetero-dimerization between SecR and the GLP-1R was demonstrated to exist and to be functionally important in receptors naturally expressed in pancreatic islets from

[1]Department of Molecular Pharmacology and Experimental Therapeutics, Mayo Clinic, Scottsdale, AZ, USA. [2]Drug Discovery Biology, Monash Institute of Pharmaceutical Sciences, Monash University, Parkville, VIC, Australia. [3]ARC Centre for Cryo-electron Microscopy of Membrane Proteins, Monash Institute of Pharmaceutical Sciences, Monash University, Parkville, VIC, Australia. ✉e-mail: denise.wootten@monash.edu; patrick.sexton@monash.edu; miller@mayo.edu

wild-type (WT) mice, comparing this to GLP-1R knockout mice[2]. SecR/GLP-1R hetero-dimeric receptor complexes were relevant for calcium responses and glucose-dependent insulin responses to secretin[2].

While high resolution structures of all class B GPCRs have now been solved, typically using cryo-electron microscopy (cryo-EM)[10–17], all of these have been monomeric forms of the receptor. Homo-dimerization of the SecR has been demonstrated by saturation biolu-minescence resonance energy transfer (BRET) spectroscopy[5], single molecule fluorescence resonance energy transfer (FRET) imaging[8], spatial intensity distribution analysis[9], and fluorescence intensity fluctuation analysis[18], with these present approximately 70 percent of the time[8,9]. However, gaining direct high-resolution insights into the structure of such complexes has been quite challenging. This likely reflects a dynamic process of association and dissociation of the pro-tomers, with difficulty in capturing a stable receptor homo-dimer for direct structural elucidation, particularly in the context of an active complex with G protein[19].

In this work, our goal is to gain a better understanding of the functionally important homo-dimeric state of SecR. For this, we utilize pharmacological, biochemical, and biophysical techniques, applied to receptor-bearing intact cells and cell membranes, to gain insights into potential differences in the binding of natural secretin at homo-dimeric SecR complexes relative to a monomeric form of this receptor. These studies utilize the WT receptor that is predominantly in a homo-dimeric complex in the cell membrane, and a well-characterized SecR construct in which mutation of two lipid-facing residues within TM4 (SecR(G264A,I268A)) disrupts dimerization[5], allowing study of mono-meric SecR behavior.

## Results

### Secretin binding and function at WT and mutant SecR in intact cells

For these studies, we utilized receptor-bearing intact cells whenever possible, where the natural high levels of endogenous guanine nucleotides can support normal G protein cycle events. For this, we prepared CHO-K1 cell lines stably expressing WT or non-dimerizing human SecR (SecR(G264A,I268A)) at similar receptor densities (Table 1). In this environment, we observed a higher potency stimula-tion of cAMP accumulation by secretin at the WT SecR relative to SecR(G264A,I268A) (Table 1 and Supplementary Fig. 1), similar to our previous report using receptors transiently expressed in COS-1 cells[5]. Under these conditions, secretin binding affinity was not different in the two cell lines.

### Spatial approximation of N-terminal secretin residues with non-dimerizing mutant SecR in intact cells

We performed cysteine trapping in intact cells to elucidate spatial approximations between distinct positions in the N-terminal activation domain of the secretin peptide and residues within the external sur-face of SecR(G264A,I268A). To achieve this, cysteines were incorpo-rated into distinct positions of interest in the peptide ligand and this receptor construct. We previously utilized this approach to study the WT SecR, probing with analogous agonist and antagonist peptide ligands that incorporated cysteine residues in positions 2, 5, 6, and 7, recognizing the critical importance of the peptide N-terminal region for agonist activity[11,20,21]. Indeed, differences were revealed in the spa-tial approximation with WT SecR of analogous position probes incor-porated into agonists and antagonists[11]. Here, we probe the spatial approximation of the same series of agonist probes utilizing identical methodology with the non-dimerizing TM4 mutant SecR construct to interrogate whether the ability of SecR to dimerize alters the engage-ment of peptide residues of the ligand activation domain.

We generated non-dimerizing SecR TM4 mutant constructs (SecR(G264A,I268A)) that also incorporated individual cysteine resi-dues in the regions of the receptor facing the ectodomain. This included residues in the N-terminal stalk region and extracellular loop regions, ECL1, ECL2, and ECL3 (analogous to the cysteine mutants that we previously studied for WT SecR). These 75 receptor constructs were characterized for expression on the surface of cells and for their ability to bind secretin (Supplementary Table 1). We performed the cysteine trapping experiments in an analogous manner to previously reported for the WT SecR[21]. Representative autoradiographs illustrating results are shown in Fig. 1, with additional autoradiographs shown in Sup-plementary Fig. 3, and quantitative data shown in Supplementary Tables 2 and 3. For each region, we included the most efficiently labeled cysteine mutant from the WT SecR as a positive control to mark the position of the receptor. Of note, no residue exhibited robust labeling above background in the N-terminal stalk region or in ECL1 (Supplementary Fig. 3). This was similar to previous observations at WT SecR[21]. There were clear differences in the cysteine trapping pat-terns of the TM4 mutant SecR relative to that previously observed with WT SecR[20,21].

Residues within SecR(G264A,I268A) with significant labeling above background were determined using one-way analysis of var-iance (ANOVA) with Dunnett's post-test, with $P < 0.001$ considered to be significant (Supplementary Table 3). These are marked in red in Fig. 1A–C. All four probes had significant sites of labeling within ECL2 of the monomeric SecR, whereas the most dominant site(s) of labeling

**Table 1 | Binding and biological activity parameters of secretin probes on CHO cell lines expressing human WT and mutant SecR**

| | pKi | P values vs Sec | $B_{max} \times 10^3$ sites/cells | P values vs Sec | pEC$_{50}$ | P values vs Sec |
|---|---|---|---|---|---|---|
| SecR WT | | | | | | |
| Sec | 8.5 ± 0.1 | | 68 ± 5 | | 11.3 ± 0.1 | |
| Alexa[488]-Sec | 7.6 ± 0.1** | 0.001 | 66 ± 7 | 0.762 | 10.2 ± 0.2** | 0.001 |
| (Lys[13]-Alexa[488])Sec | 8.3 ± 0.1 | 0.30 | 67 ± 6 | 0.993 | 10.6 ± 0.2* | 0.027 |
| (Lys[22]-Alexa[488])Sec | 8.2 ± 0.1 | 0.098 | 68 ± 7 | 0.998 | 10.8 ± 0.1 | 0.10 |
| Sec-Gly[28]-(Cys[29]-Alexa[488]) | 8.1 ± 0.1 | 0.07 | 66 ± 8 | 0.654 | 11.1 ± 0.1 | 0.93 |
| SecR(G264A,I268A) | | | | | | |
| Sec | 8.6 ± 0.1 | | 60 ± 6 | | 10.6 ± 0.1# | 0.029 |
| Alexa[488]-Sec | 7.6 ± 0.2** | 0.004 | 62 ± 4 | 0.926 | 9.4 ± 0.3** | 0.002 |
| (Lys[13]-Alexa[488])Sec | 8.3 ± 0.2 | 0.47 | 61 ± 5 | 0.981 | 9.9 ± 0.3 | 0.10 |
| (Lys[22]-Alexa[488])Sec | 8.1 ± 0.1 | 0.14 | 61 ± 9 | 0.996 | 9.9 ± 0.1 | 0.069 |
| Sec-Gly[28]-(Cys[29]-Alexa[488]) | 8.6 ± 0.1 | >0.99 | 61 ± 8 | 0.972 | 10.5 ± 0.2 | 0.98 |

Values are expressed in means ± S.E.M. of four independent experiments performed in duplicate, and significant differences were tested using one-way ANOVA with Dunnett's post-test.
*$P < 0.05$, **$P < 0.01$, significantly different from natural secretin peptide at the same receptor construct; #$P < 0.05$, significantly different from the same condition at the SecR WT construct.

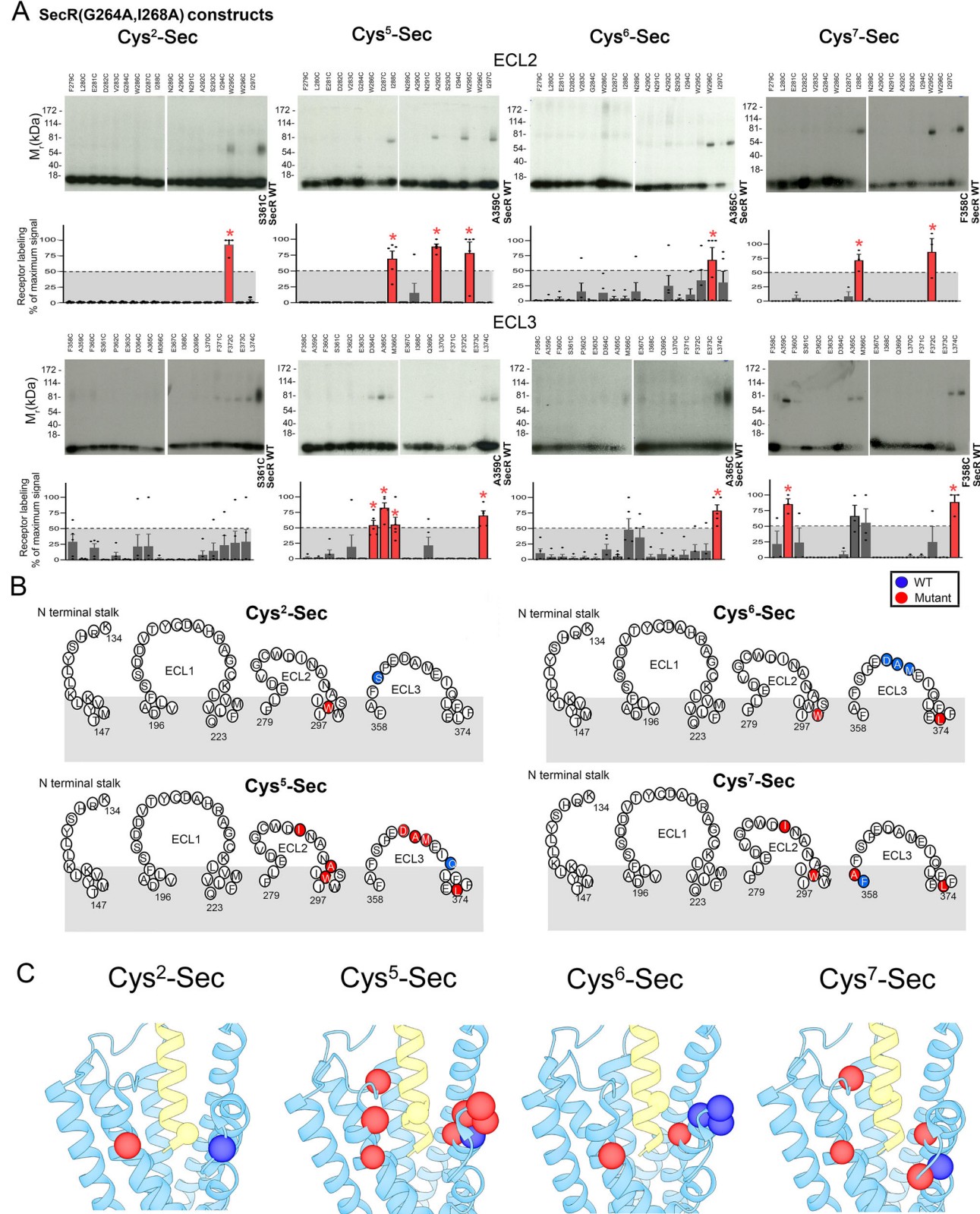

WT SecR was always in ECL3. The Cys 5, 6, and 7 probes labeled ECL3 residues both in WT and non-dimerizing SecR, however the monomeric form of the receptor was predominantly labeled at the region of this loop closest to TM7, while the WT SecR exhibited more distributed labeling across ECL3, in a probe-dependent manner. These changes in spatial approximation have been illustrated schematically as mapped onto our published cryo-EM structure of monomeric SecR in complex

with human secretin and G protein[11] in Fig. 1C. These data indicate that the ability of the receptor to form dimers is associated with the peptide N-terminal region moving away from ECL2, while maintaining proximity to ECL3. No structures of dimeric complexes of this receptor or any other class B GPCR have yet been reported.

To gain additional insights into the impact of homo-dimerization on the binding of the agonist secretin peptide, we moved to membrane

**Fig. 1 | Cysteine trapping of Cys²-Sec, Cys⁵-Sec, Cys⁶-Sec, Cys⁷-Sec to cysteine mutants of non-dimerizing SecR(G264A,I268A) construct expressed on intact cells.** Shown in (**A**) are representative autoradiographs of 10% SDS-PAGE gels used to separate products of cysteine trapping of SecR mutants across ECL2 and ECL3 expressed in COS-1 cells by each noted probe. Gels were run in the absence of any reducing agent, and control receptor labeling on each gel was detected using key cysteine mutants of WT SecR. Shown also are the densitometric analysis of data from three to five independent experiments (Cys²-Sec, $n = 4$; Cys⁵-Sec and Cys⁶-Sec, $n = 5$; and Cys⁷-Sec, $n = 3$), with dots illustrating each data point. The receptor labeling signal was calculated relative to the intensity of the residue with the highest labeling using that probe across all regions. Sites of significant labeling above background were determined using one-way ANOVA with Dunnett's post-

test, with $P < 0.001$ considered to be significant (absolute values shown in Supplementary Table 3). In (**A**), these are colored red and marked with *. **B** provides a schematic representation of the labeling across all regions, showing the WT SecR residue(s) with the highest labeling intensity previously published[11,20,21] in blue, and the significantly labeled residues identified in the current work with the non-dimerizing mutant SecR(G264A,I268A) shown in red. **C** schematically illustrates these sites of cysteine trapping of the non-dimerizing mutant SecR (red spheres) along with the highest sites of covalent labeling of WT SecR previously reported[11] (blue spheres), as mapped onto our published cryo-EM structure of monomeric SecR in complex with human secretin and G protein[11] (SecR models displayed in ChimeraX version 1.6.1).

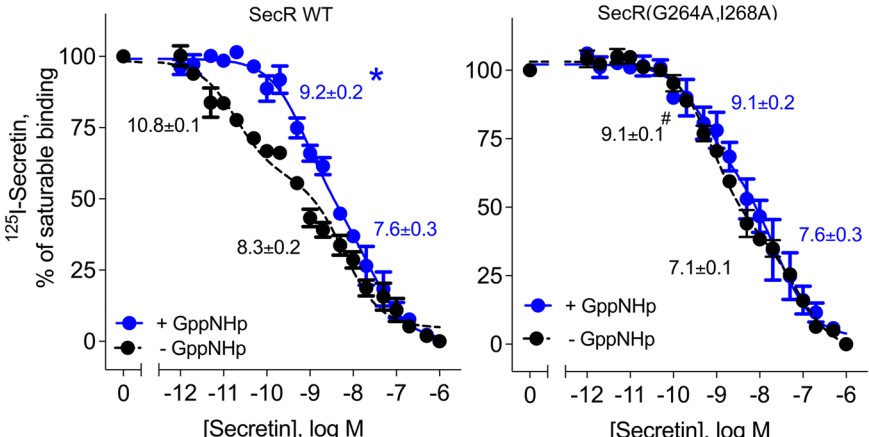

**Fig. 2 | Binding analysis of SecR constructs expressed in cell membranes.** Shown are the competition-binding data for membranes expressing WT SecR (left panel) and non-dimerizing mutant SecR (right panel) in the absence (black) or presence (blue) of 10 μM GppNHp. Values are expressed as percentages of saturable binding and displayed as means ± S.E.M. from three independent experiments performed in duplicate, and analyzed using the Mann–Whitney test. Values of Ki for high and low affinity sites are noted, with only the high affinity site in SecR WT significantly different in the presence and absence of GppNHp at this receptor ($P = 0.011$)*. The affinity of this high affinity binding site in the absence of GppNHp is also significantly different in SecR WT from that in SecR(G264A,I268A) ($P = 0.0136$)#. (all values shown in Supplementary Table 4).

preparations that also provided the opportunity to engineer the status of G protein engagement, where the isolated membranes are relatively depleted in guanine nucleotides (a condition expected to favor longer duration G protein interactions than might exist in intact cells). Treating such membranes with a non-hydrolyzable GTP analog, guanosine-5′-[(β,γ)-imido]triphosphate (GppNHp), can irreversibly uncouple the G protein from the receptor after a single cycle of guanine nucleotide exchange (uncoupled state). Nucleotide treated and untreated membranes were prepared from the cell lines expressing WT SecR and SecR(G264A,I268A) described above. These studies also utilized a series of secretin analogs that we had previously developed, incorporating an alexa⁴⁸⁸ fluorophore across the secretin peptide pharmacophore, in positions −1, 13, 22, and 29[22], as fluorescence indicators of the local microenvironment. The positions of the fluorophores in these probes are highlighted using the active SecR structure[11], illustrating that both N-terminal and C-terminal probes are deeply embedded in the helical bundle and ECD, respectively (Supplementary Fig. 4), while the position 13 and 22 probes are located within interfaces at the junction of the transmembrane domain core and the ECD.

**Fluorescent secretin probe characterization**
To further validate the utility of these probes, their binding and signaling at the stable WT and mutant SecR-expressing CHO cell lines were characterized (Table 1 and Supplementary Fig. 1). Supplementary Fig. 2 shows that these probes bound saturably to the surface of the receptor-bearing cells. These results were analogous to our previously reported data[22]. The N-terminal probe (position -1) exhibited lower

affinity and lower potency in cAMP accumulation assays, than natural secretin, consistent with the known importance of the peptide N terminus for binding and biological activity[22]. This was true for both WT SecR and SecR(G264A,I268A). The position 13 probe also exhibited lower potency in eliciting cAMP accumulation than natural secretin at the WT SecR, while binding with equivalent affinity. The position 22 and 29 probes bound and signaled normally at both WT SecR and SecR(G264A,I268A) (Table 1 and Supplementary Fig. 1).

**Secretin binding at WT and monomeric mutant SecR in membranes**
The characteristics of SecR binding in cell membrane preparations were different to those in intact cells. Shown in Fig. 2 (Supplementary Table 4), both high affinity and low affinity states were observed at the WT receptor. Uncoupling the G protein using GppNHp treatment significantly reduced the affinity of the high affinity binding, while the low affinity binding affinity was not different. Under these conditions, binding to the high affinity site represented approximately 45% of total binding (Supplementary Table 4). Of note, analogous studies with the non-dimerizing SecR mutant, SecR(G264A,I268A), demonstrated binding affinities similar to those observed for the WT SecR membranes treated with GppNHp, however, binding to the mutant SecR was not significantly affected by treatment with GppNHp (Fig. 2 and Supplementary Table 4.

**Secretin fluorescent probe association and dissociation kinetics**
The fluorescent secretin probes used in the biophysical experiments were also used to evaluate peptide binding and dissociation kinetics to

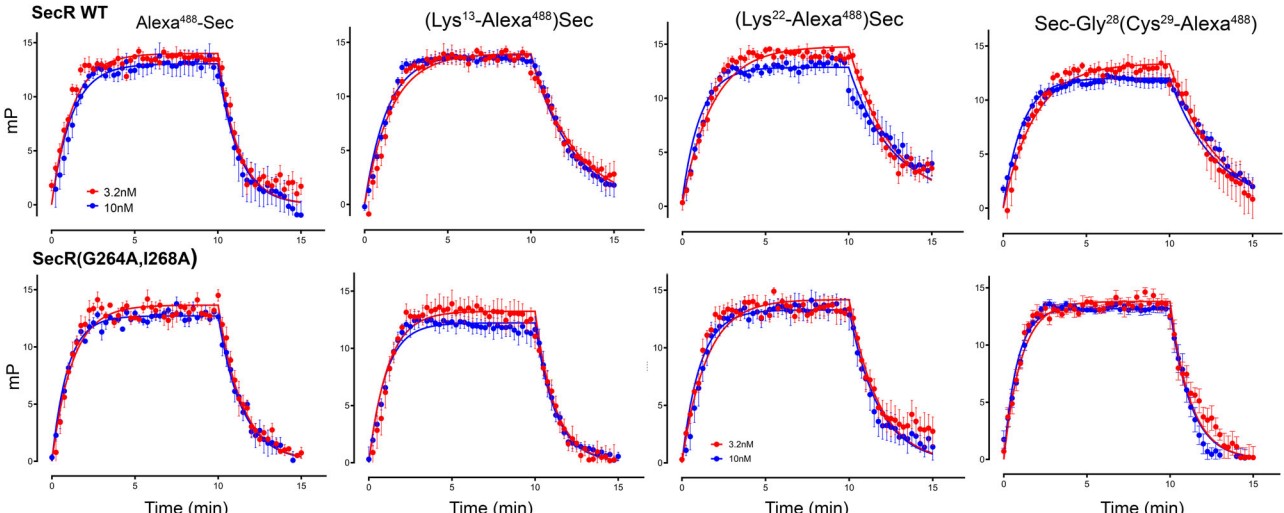

**Fig. 3 | Kinetic binding profiles of Alexa[488] secretin probes in WT and mutant secretin receptors.** Shown are the kinetic receptor binding profiles of fluorescent probes using fluorescence polarization at membranes bearing WT SecR (top row) and mutant SecR (bottom row). N-terminal probe (Alexa[488]-Sec) has a faster off rate compared to the other probes for WT SecR, while position 13 and C-terminal probes exhibited faster off rates for mutant SecR ($P = 0.0133$ and $0.0321$, respectively, determined using one-way ANOVA with Tukey post-test). Values are expressed as means ± S.E.M. from five independent experiments performed in triplicate. Absolute values and statistical analysis are shown in Supplementary Table 5.

the WT and mutant SecR-expressing cell membranes (Fig. 3 and Supplementary Table 5), yielding similar values for the human WT SecR to those previously reported for rat WT SecR[22]. Because the fluorophore at the N terminus of the peptide interfered with normal binding, its binding affinity for WT SecR was lower than that of the other probes. This is consistent with its more rapid dissociation rate compared to that of the other probes ($P = 0.022$) (Supplementary Table 5). Comparison of the kinetics at WT SecR relative to non-dimerizing SecR revealed similar on-rates and a tendency toward faster off-rates for the monomeric receptors, reaching statistical significance for the position 13 and 29 probes ($P = 0.0133$ and $0.0321$, respectively) (Supplementary Table 5).

### Secretin fluorescent probe quenching

The data for quenching of the fluorescence for these probes provided insights into their accessibility to the hydrophilic quenching reagent, KI (Fig. 4, Supplementary Table 6). The ease of quenching correlates with the slope of the lines. WT SecR with the fluorophore at the N terminus exhibited the steepest curves, consistent with ease of quenching and, therefore, accessibility ($P = 0.002$). This was true both in the control state ($P = 0.002$) and in the presence of GppNHp ($P = 0.0254$). Interestingly, this was not evident in the non-dimerizing mutant constructs, with all probes exhibiting similar degrees of quenching. Of note, in the WT SecR, uncoupling G proteins with GppNHp significantly reduced the quenching constants for the mid-region probes in positions 13 and 22 ($P = 0.009$ and $0.003$, respectively). This suggests that the mid-region of secretin is more buried and less accessible to KI quenching when uncoupled from its G protein than in its control state.

### Secretin fluorescent probe anisotropy

Anisotropy measurements reflect the rotational motion of fluorophores, with lower anisotropy often correlating with higher mobility. The anisotropy is also dependent on temperature, with that observed at higher temperature having lower anisotropy values (higher mobility). The anisotropy data (Fig. 5 and Table 2) were consistent with the implications of the fluorescence quenching (Fig. 4 and Supplementary Table 6). All the WT SecR probes exhibited similar baseline anisotropy and all the non-dimerizing mutant probes were similar to each other as well, although the levels of anisotropy were greater for the position −1,

13, and 22 probes at the mutant receptor than at the WT receptor. This higher anisotropy suggests lower mobility of the probes bound to the monomeric form of SecR. For WT SecR, the probes in positions 13 and 22 exhibited higher anisotropy after treatment with GppNHp than the control state. This was true both at 20 and at 37 °C. This was not observed for any of the probes at the non-dimerizing mutant SecR.

## Discussion

Homo-dimeric complexes of SecR are the predominant form of WT receptor expressed on the plasma membrane, representing approximately 70% of receptors present[8,9], with dimerization facilitating high potency responses to natural agonist ligand[23]. Indeed, the functional importance of dimerization was further validated in the current studies, in intact cells that have a normal complement of guanine nucleotides to support the full cycle of G protein association-dissociation events.

It is noteworthy that the apparent affinity of secretin in intact cells was not different between WT SecR and the SecR TM4 mutant that disrupts the dimer interface. However, in membrane binding studies where G protein interaction and turnover can be engineered through either depletion of nucleotide to stabilize the ternary complex, or through G protein uncoupling by treatment with a non-hydrolyzable analog of GTP, GppNHp, two distinct populations of binding sites were observed. The high affinity site was more prevalent in the control WT SecR membranes than when these were treated with GppNHp, while the low affinity binding site was unaffected. In contrast, the affinities of the two sites in the mutant SecR(G264A,I268A) membranes were not different from the affinities observed in the WT SecR membranes treated with GppNHp. This suggests that the high potency cAMP response at WT SecR is likely mediated by this high affinity binding, but that in intact cells with high endogenous guanine nucleotide this is a transient, less dominant, state of the receptor.

It has been highly challenging to structurally capture the dimeric state of SecR and all other members of this receptor subfamily. While high resolution monomeric structures exist for all class B1 GPCRs[10–17], no dimeric structure has yet been solved. In the current work, we provide evidence that dimerization alters peptide binding and dynamics that likely underlies the functional differences observed in cells.

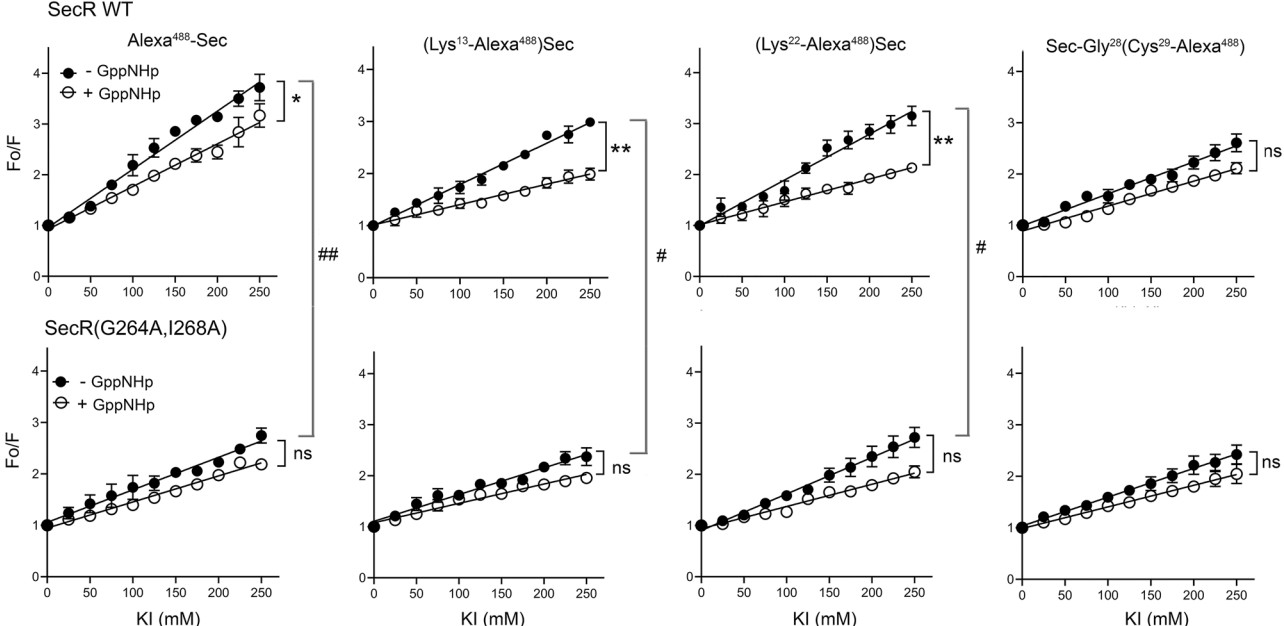

**Fig. 4 | Fluorescence quenching of receptor-bound probes in active and G protein-uncoupled states.** Shown are the KI Stern–Volmer collisional quenching patterns of receptor-bound probes at WT SecR (top panel) and non-dimerizing mutant SecR (bottom panel) in the absence (closed circles) or presence (open circles) of GppNHp. Values are expressed as means ± S.E.M. from 3 independent experiments, and analyzed using one-way ANOVA with Tukey post-test. Shown here are comparisons reaching: * or #: $P < 0.05$; ** or ##: $P < 0.01$. Absolute values and statistical analysis are shown in Supplementary Table 6.

The formation of disulfide bonds between cysteines requires both spatial proximity and appropriate geometry[24], making cysteine trapping (crosslinking) experiments a sensitive readout of the location and dynamics of the N-terminal cysteine-substituted secretin peptide analogs when binding to the receptor in intact cells. When SecRs were restricted to the monomeric state, the secretin probes formed interactions across ECL2 and ECL3, consistent with the enclosed TM peptide binding pocket observed in the active, G protein-coupled SecR structure isolated in the monomeric state (Supplementary Fig. 4). In contrast, the highest sites of cysteine cross-links were mainly within ECL3 at the "WT" SecR, indicating that the peptide N-terminal region moves away from ECL2, but remains proximal to ECL3, when in a "dimer competent" state.

Previous analysis of the conformational dynamics of the active, monomeric, SecR cryo-EM structure[11] indicated that binding of the peptide N terminus within the TM core was stable, with interactions with ECL2 and ECL3 maintained throughout the trajectory of each of the principal components of motion (Supplementary Movie 1), further supporting the observed cysteine cross-links in the non-dimerizing SecR construct. Intriguingly, in select structures of other class B1 GPCRs bound to different peptides, dynamic changes in the conformations of ECL3 and peptide engagement have been observed that would be consistent with the pattern of cross-linking in the WT SecR. For example, structural analysis of GLP-1R binding to the agonist peptides, GLP-1, semaglutide, and taspoglutide, revealed comparable high-resolution consensus structures with a closed TM conformation bound to each of the peptides[25], similar to that observed for the secretin-bound SecR, but distinct conformational dynamics. Analysis of those conformational dynamics, derived from the cryo-EM data, revealed that ECL3 underwent outward movement, in the complexes with semaglutide or taspoglutide, with a parallel shift in the peptide N terminus away from ECL2 while maintaining interaction with ECL3[25]. This was postulated to be related to the mechanism of peptide binding and unbinding, and later to the efficiency of G protein activation of highly efficacious, lower affinity, peptide agonists[26], where partial disengagement of the peptide N-terminal activation domain was speculated to be linked to G protein activation and release. The pattern of cysteine cross-linking at the WT SecR is consistent with the more open conformation and position of the peptide N terminus observed in the conformational analysis of the semaglutide and taspoglutide bound GLP-1R complexes (Fig. 6), consistent with SecR dimerization facilitating increased conformational dynamics and more efficient G protein signaling. This hypothesis was further supported by our biophysical studies.

Attachment of the alexa[488] fluorophore to the −1 position, unsurprisingly lead to reduced affinity and potency in cAMP production at the WT receptor, which was associated with increased KI quenching, consistent with a reduction in binding deep into the receptor core, an event likely required to support efficacious G protein recruitment. Moreover, there was limited GppNHp sensitivity, either in fluorescence quenching or anisotropy measurements. Intriguingly, despite also having reduced affinity and potency at the non-dimerizing mutant, the fluorophore was more buried and less conformationally dynamic than at the WT receptor. This indicates that the monomeric SecR has more stable binding and is less conformationally dynamic even when occupied with this lower potency probe.

The mid-region probes that had affinity and potency similar to the parental secretin peptide provided important insights into receptor behavior. Both the position 13 and 22 probes demonstrated greater solvent exposure, dynamics and nucleotide sensitivity at WT SecR, relative to the monomeric receptor. In contrast, the position 29 probe had similar biophysical characteristics at WT and mutant receptor. Collectively, these data are indicative of a similar, relatively buried, interaction of the peptide C terminus with the receptor ECD that is not influenced by G protein binding and activation, consistent with current two-domain models of peptide binding to class B1 GPCRs. In those models, the initial binding of peptides occurs with the ECD, allowing subsequent positioning of the peptide N terminus to facilitate productive engagement of the N-terminal activation domain with the receptor core[27]. The KI quenching and anisotropy data of the mid-region probes demonstrates that this region of the peptide is dynamic in the WT receptor, in a G protein binding-dependent manner.

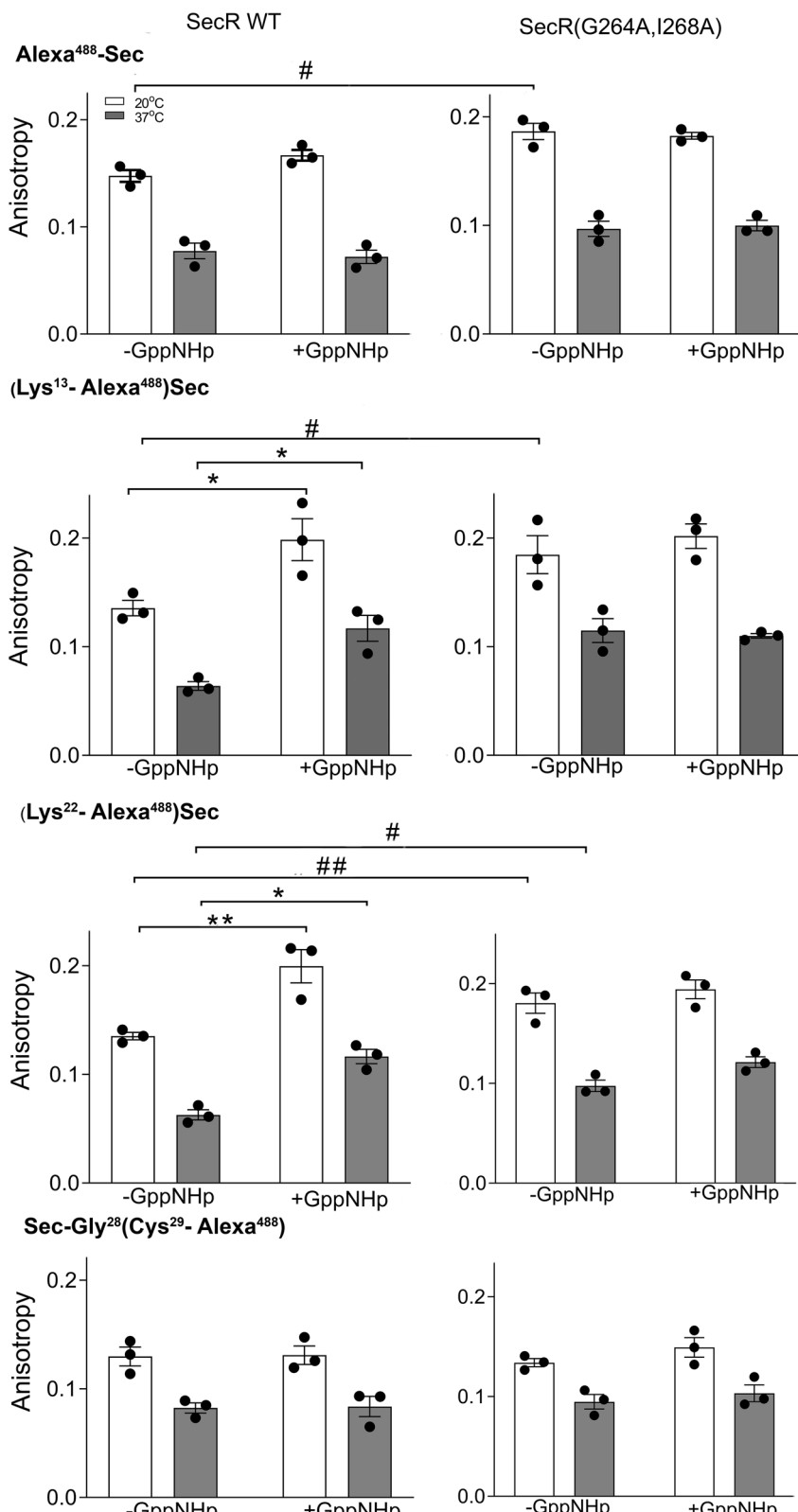

**Fig. 5 | Fluorescence anisotropy of receptor-bound probes in active and G protein-uncoupled states.** Shown are the steady state anisotropy values of secretin fluorescent probes bound to WT SecR (left column) and non-dimerizing mutant SecR (right column) in the presence or absence of GppNHp, a non hydrolyzable analog of GTP. The values are shown as means ± S.E.M. from three independent experiments, and were analyzed using one-way ANOVA with Tukey post-test. Differences between active and G protein-uncoupled states (+GppNHp) are noted, *$P < 0.05$. Differences between the active states of WT SecR and mutant SecR are marked, #$P < 0.05$. Absolute values and statistical analysis are shown in Table 2.

**Table 2 | Anisotropy data for secretin probes bound to WT and mutant SecR**

| Probes | Control | | +GppNHp | | Comparison Control vs +GppNHp, P values | | Comparison WT vs mutant, P values | |
|---|---|---|---|---|---|---|---|---|
| | 20 °C | 37 °C | 20 °C | 37 °C | 20 °C | 37 °C | 20 °C | 37 °C |
| SecR WT | | | | | | | | |
| Alexa[488]-Sec | 0.15 ± 0.01 | 0.08 ± 0.01 | 0.17 ± 0.01 | 0.07 ± 0.01 | 0.190 | 0.918 | | |
| (Lys[13]-Alexa[488])-Sec | 0.14 ± 0.01 | 0.06 ± 0.01 | 0.20 ± 0.02* | 0.12 ± 0.01* | 0.046 | 0.044 | | |
| (Lys[22]-Alexa[488])-Sec | 0.14 ± 0.01 | 0.06 ± 0.01 | 0.20 ± 0.02** | 0.12 ± 0.01* | 0.004 | 0.011 | | |
| Sec-Gly[28]-(Cys[29]-Alexa[488]) | 0.13 ± 0.01 | 0.08 ± 0.01 | 0.13 ± 0.01 | 0.08 ± 0.01 | 0.999 | 0.999 | | |
| SecR(G264A, I268A) | | | | | | | | |
| Alexa[488]-Sec | 0.19 ± 0.01[#] | 0.1 ± 0.01 | 0.18 ± 0.01 | 0.1 ± 0.01 | 0.960 | 0.982 | 0.0163 | 0.266 |
| (Lys[13]-Alexa[488])-Sec | 0.18 ± 0.02[#] | 0.11 ± 0.01 | 0.20 ± 0.01 | 0.11 ± 0.01 | 0.749 | 0.959 | 0.0477 | 0.102 |
| (Lys[22]-Alexa[488])-Sec | 0.18 ± 0.01[##] | 0.10 ± 0.01[#] | 0.19 ± 0.01 | 0.12 ± 0.01 | 0.635 | 0.229 | 0.005 | 0.0230 |
| Sec-Gly[28]-(Cys[29]-Alexa[488]) | 0.13 ± 0.01 | 0.09 ± 0.01 | 0.15 ± 0.01 | 0.1 ± 0.01 | 0.331 | 0.748 | 0.968 | 0.565 |

Values are expressed as means ± S.E.M. from three independent sets of observations analyzed using one-way ANOVA with Tukey post-test.
*$P < 0.05$ and **$P < 0.01$, significantly different from control at the same receptor construct, [#]$P < 0.05$ and [##]$P < 0.01$, significantly different from analogous condition at WT SecR.

Decoupling of the G protein reduces quenching and increases anisotropy, consistent with stabilization of binding in this state. In contrast, the mid-region probes bound to the monomeric receptor are less conformationally dynamic and are not further altered by G protein decoupling. Insight into the nature of the conformational dynamics underlying the behavior of the mid-region probes can be inferred from the observed conformational dynamics of the secretin-bound SecR structure, where a twisting of the ECD, and associated peptide, relative to the core is seen (Supplementary Movie 1). Furthermore, this is consistent with a model where the dynamics of the ECD and mid-region of the peptide helps to drive partial disengagement of the peptide N terminus from the deep binding pose. This is likely correlated with the G protein activation event, which is also consistent with the cysteine cross-linking data discussed above. In the monomeric state, the extent of the dynamic motions of the peptide mid-region is likely limited by greater stability of interactions of the peptide N terminus with ECL2 and ECL3.

Collectively, our data provides support for a model whereby the ability of SecR to form dimers alters the dynamics of the receptor, in a G protein-dependent manner (Fig. 7). Precedent for an associated membrane protein, RAMP2, to affect class B GPCR flexibility and G protein coupling has previously been observed[28]. In the current study, the increased dynamics induced by the SecR homo-dimer allow engagement and disengagement of the peptide N-terminal activation domain with the deep pose required for G protein recruitment. This leads to increased rates of G protein activation and release, which is reflected in the binding affinity in membranes and potency in whole cell second messenger signaling. The ability to dimerize is associated with increased dynamics of the peptide mid-region linked to rotational movement of the ECD relative to the core, facilitating partial N-terminal peptide disengagement in parallel with the increased dynamics of ECL3. Overall, our study advances our understanding of class B1 GPCRs and provides a structural rationale for how dimerization is functionally important for high potency signaling.

## Methods
### Materials
Dulbecco's Modified Eagle's medium (DMEM), Hams F-12 medium and soybean trypsin inhibitor were from Invitrogen (Carlsbad, CA). Polyethylenimine was from Polysciences (Warrington, PA). Fetal Clone II was from Hyclone laboratories (Logan, UT) and other tissue culture supplements were purchased from Life Technologies (Carlsbad, CA). Guanosine-5'-[(β,γ)-imido]triphosphate (GppNHp) was from Millipore-Sigma (St. Louis, MO). Bovine serum albumin was from Serologicals Corp. (Norcross, GA). All other reagents were analytical grade.

Na[125]I used for peptide radioiodination and HTRF cAMP Gs dynamic kit were from Perkin Elmer (Boston, MA). Iodobeads were from Pierce Chemicals (Rockford, IL), Novex precast gels were from Life Technologies (Carlsbad, CA), and Prosieve gel markers were from Lonza (Rockland, ME).

### Peptides
As described previously, four cysteine-containing secretin agonist peptide probes were designed, incorporating cysteines for disulfide trapping into positions 2, 5, 6 and 7 of human secretin(1-27) (labeled as Cys[2]-, Cys[5]-, Cys[6]-, and Cys[7]-agonist probes)[20,21]. Tyr was incorporated into position 10 of these probes for radioiodination. All peptides were synthesized in our laboratory using manual solid phase techniques, purified by reversed-phase HPLC, and characterized by mass spectrometry, as described previously[21]. Fluorescent secretin probes were prepared by incorporating Alexa[488] into four different positions along the peptide, -1, 13, 22, and 29: N-terminus (-1), Lys[13], Lys[22], and Gly[28]-Cys[29] (C-terminus) of natural secretin(1-27), as previously described[22].

### Peptide radioiodination
For cysteine trapping (crosslinking) studies, Cys[2]-, Cys[5]-, Cys[6]- and Cys[7]-[Tyr[10]]secretin(1-27) agonist probes and Tyr[10]-secretin(1-27) were radioiodinated using oxidative techniques[29]. In short, 15 µg of each peptide were incubated with 1 mCi of Na[125]I for 15 s in borate buffer (pH 9.0) using solid-state N-chlorobenzene sulfonamide (iodination beads), and radioactive peptide products were separated and purified using reversed-phase HPLC, to yield a mono-iodinated peak with a specific radioactivity of 2000 Ci/mmol.

### Receptor mutagenesis
Modified secretin receptor constructs were prepared by mutagenesis of human SecR(G264A,I268A) by introducing cysteine residues to replace each natural residue in the N-terminal stalk region, (extracellular loop 1) ECL1, ECL2, and ECL3 regions of the receptor[21]. Mutagenesis was performed using the QuikChange site-directed mutagenesis kit from Stratagene (La Jolla, CA), with the products verified by direct DNA sequencing.

### Cell culture and transfection
COS-1 cells (American Type Culture Collection, Manassas, VA) or CHO-K1 cell lines (American Type Culture Collection, Manassas, VA) were used for studies involving receptor binding characterization, cell surface expression and cysteine cross linking. COS-1 cells were cultured on tissue culture plasticware in DMEM supplemented with 5% Fetal Clone II at 37 °C in an environment containing 5% $CO_2$. Chinese

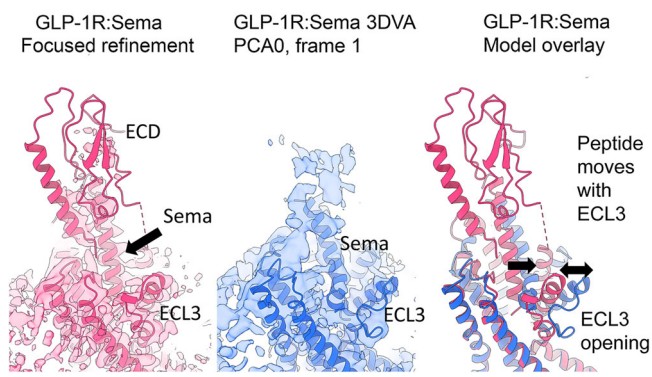

**Fig. 6 | Conformational dynamics of a class B1 GPCR.** Conformational dynamics of the active, Gs protein-coupled, GLP-1 receptor bound with agonist analog, semaglutide, provide a potential model for the shifts in cysteine cross-linking patterns of WT and non-dimerizing mutant SecR. This shows cryo-EM data of semaglutide bound to GLP-1R[25], showing focused refinement map and model (left, pink) and map and model of principal component 0 (PCA0) of the CryoSPARC 3D variability analysis (3DVA) (middle, blue), as well as an overlay of both models (right). In the closed conformation (left panel) the semaglutide peptide is proximal to ECL2 and ECL3, similar to the conformation of the active, monomeric SecR-secretin structure and pattern of cysteine cross-linking at the non-dimerizing SecR suggested by the current work. Outward movement of ECL3 is accompanied by a translation of the semaglutide peptide (middle and right panels) that moves away from ECL2, but remains proximal to ECL3, consistent with the pattern of cysteine cross-linking observed at the WT (dimer competent) SecR.

Hamster Ovary (CHO) cells were maintained in Ham's F-12 medium supplemented with 5% Fetal Clone II, and passaged approximately twice per week. CHO cell lines with matching levels of expression of WT and mutant SecR(G264A,I268A) were prepared by transfection using polyethylenimine (PEI) followed by cloning using limiting dilution[30].

### Secretin receptor immunostaining
Cell surface expression of secretin receptor cysteine mutant constructs in COS-1 cells was monitored by immunostaining using secretin receptor-specific polyclonal antiserum[11]. This antiserum was raised against a peptide antigen representing amino acids hSecR(52–66). Receptor-bearing cells were grown on glass coverslips in six-well plates for 24 h and washed once with PBS, pH 7.4, followed by fixation with 2% paraformaldehyde (Electron Microscopy Sciences, Hatfield, PA) in PBS for 15 min. The coverslips were then washed with PBS and incubated with this antiserum (1:400 in PBS with 1% normal goat serum) in a humidified chamber for 1 h at room temperature. Coverslips were washed further with PBS containing 1% normal goat serum and incubated for 1 h with Alexa[488]-conjugated anti-rabbit IgG secondary antibody (1:250) (Molecular Probes, Eugene, OR). After the incubation, cells were washed and mounted on microscope slides with Vectashield mounting medium (Vector Laboratories, Burlingame, CA). Cell surface fluorescence was collected using a Zeiss inverted microscope (×40 objective) controlled by QED InVivo software (Media Cybernetics, Bethesda, MD). Cell surface fluorescence was quantified using ImageJ software (National Institutes of Health, Bethesda, MD).

### Membrane preparation
Receptor-expressing membranes were isolated from CHO-SecR cells using discontinuous sucrose density gradient centrifugation, as previously described[31]. Membranes were stored at −80 °C.

### Receptor binding assays
Radioligand binding assays were performed using cells in 24-well tissue culture plates or membranes in suspension. In brief, transfected COS-1 cells, receptor-expressing CHO cell lines, or receptor-bearing

membranes were incubated with a constant amount of radioligand, [[125]I-Tyr[10]]secretin(1-27) (~11.2 pM, approximately 10,000 cpm), in the absence and presence of increasing concentrations of secretin peptide or fluorescent secretin probes (ranging from 0 to 1 μM) for 1 h at room temperature in Krebs-Ringers/HEPES (KRH) medium (25 mM HEPES, pH 7.4, 104 mM NaCl, 5 mM KCl, 2 mM CaCl$_2$, 1 mM KH$_2$PO$_4$, 1.2 mM MgSO$_4$) containing 0.01% soybean trypsin inhibitor and 0.2% bovine serum albumin. After the incubation, cells were allowed to settle and were washed twice with ice-cold KRH medium. The cell pellets were then lysed with 0.5 M NaOH before quantifying radioactivity. For membrane-bound receptor binding, free radioligand was separated from membrane-bound ligand by centrifugation at 20,000 × $g$ for 5 min, with the pellet washed twice before counting. Bound radioligand was quantified using a Berthold γ-spectrometer (Oak Ridge, TN) with 70% counting efficiency. Non-saturable binding was determined in the presence of 1 μM unlabeled secretin and represented <15% of total radioligand bound. Saturable binding data and binding kinetics were analyzed and plotted using the nonlinear regression analysis in Prism software version 9.2 (GraphPad, San Diego, CA). Membrane binding data were fit to one- and two-site models, with F test determining if the two-site model was significantly better than the one-site model, with $P < 0.05$ considered significant. Two-site data were utilized only when the $F$ test was significant.

### cAMP assays
Peptide-stimulated intracellular cAMP responses were quantified in receptor-expressing cells. Receptor-bearing cells were grown in clear 96-well plates to reach approximate 85% confluence[2]. Cells were washed with PBS, 7.4, and then stimulated with increasing concentrations of secretin or secretin probes (0 to 0.1 μM) in KRH medium, pH 7.4 supplemented with 0.1% bacitracin and 1 mM 3-isobutyl-1-methylxanthine (Millipore-Sigma, Burlington, MA) for 30 min at 37 °C. After incubation, cells were lysed with 6% ice cold perchloric acid with vigorous shaking for 15 min at room temperature, then the pH was adjusted to pH 6.0 with 30% KHCO$_3$. Aliquots of the cell lysates were used to quantify cAMP levels in 384-well Optiplates using HTRF cAMP Gs dynamic kit (Perkin-Elmer, Boston, MA) following the manufacturer's instructions, and time-resolved fluorescence was measured using a PheraSTAR FSX (BMG LabTech Inc., Cary, NC). cAMP responses were measured, with values calculated based on a cAMP standard curve and data plotted using non-linear regression with three parameter curve fitting in Prism 9.2.

### Cysteine trapping
Cysteine trapping is a technique in which cysteine residues incorporated into distinct positions within a peptide ligand and within its receptor are allowed to form disulfide bonds when two free cysteines are in proximity to each other with necessary distance and geometry[24]. As noted above, this was performed with secretin peptides incorporating cysteines in positions 2, 5, 6 and 7[20,21], and with a series of SecR mutants incorporating cysteines in all extracellular domains (75 distinct constructs in a TM4 mutant non-dimerizing SecR background). COS-1 cells were grown to approximate 85% confluence in 24-well tissue culture plates, and transfected with cysteine mutant receptor constructs using the PEI method[21]. Assays were started by washing the cells with DMEM containing 5% Fetal Clone II before being incubated with 200 μl radiolabeled cysteine-containing probes (approximately 100,000 cpm per well) for 1.5 h at room temperature. The assays were terminated by washing the cells with ice-cold PBS, pH 7.4, and mixing with 60 μl SDS Laemmli sample buffer not containing dithiothreitol (DTT). Lysates were run on 10% SDS-polyacrylamide gels. Gels were then dried, and bands of interest were visualized by autoradiography, and specific bands were identified, and intensities quantified using ImageJ. Sites having labeling above background were determined using one-way ANOVA with Dunnett's post-test, with $P < 0.001$

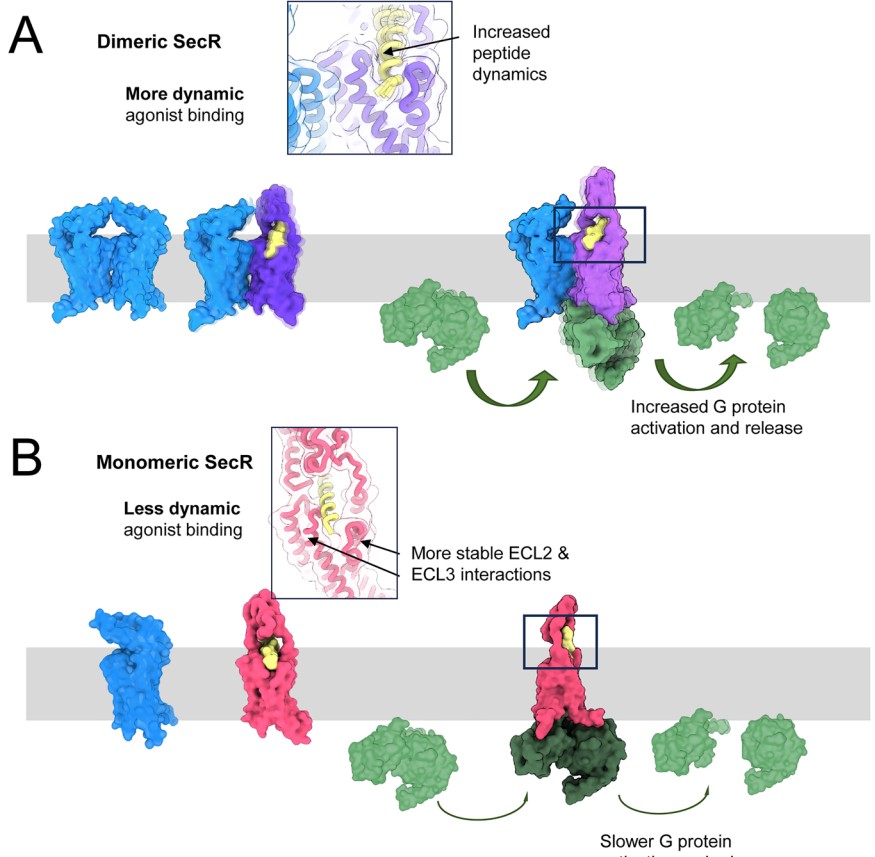

**Fig. 7 | Schematic overview of proposed state-dependent secretin receptor dynamics. A** At the WT SecR, recruitment of G protein (green) to the dimer (protomer 1, blue; protomer 2, purple) leads to increased dynamics of the receptor ECD relative to the core, and in ECL3 with partial disengagement of the peptide N-terminus from the receptor core, which remains proximal to ECL3, promoting faster G protein activation and release. **B** In the absence of the dimer, the monomeric SecR (pink) forms more stable interactions with the secretin peptide N-terminus, which remains proximal to both ECL2 and ECL3, leading to slower disengagement of the peptide following G protein coupling and a slower rate of activation and release. The secretin peptide is shown in yellow. The inset panels show the predicted higher frequency engagement of the agonist with ECL3 in one of the protomers in the dimeric state, based on cysteine-trapping experiments. In the monomer, the agonist remain proximal to both ECL2 and ECL3 for longer. The structures are artificial representations only, based on Alphafold2 predictions of the inactive secretin receptor, derived from the GPCRdb[34], as well as the published active structure (PDB: 6WZG). Inactive, dimer and agonist-bound models underwent geometry minimization and simple dynamics in Phenix version 1.20[35,36]. Structure displays were prepared using ChimeraX version 1.6.1[37].

considered to be significant. The apparent molecular weights of the receptor bands were determined by interpolation with the mobility of the appropriate ProSieve protein markers.

### Fluorescence spectrometry

Fluorescence characteristics of receptor-bound secretin alexa[488] probes (as described above, with fluorophore in positions −1, 13, 22, and 29) were determined in the presence or absence of GppNHp, a non-hydrolyzable analog of GTP, as described previously[22]. Samples were prepared by mixing the cell membrane suspension (5 μg) with 50 nM fluorescent secretin probe in the absence or presence of 10 μM GppNHp for 20 min at room temperature in KRH medium, pH 7.4. Incubations were terminated by separating receptor-bound ligand from free ligand using centrifugation at $25,000 \times g$ for 10 min at 4 °C. The sample pellet was washed again with ice-cold KRH medium and resuspended in KRH medium for fluorescence measurements. The samples were excited at 481 nm and the emission was collected at 521 nm, with a bandwidth of 5 nm at an integration rate of 0.5 nm/s, and fluorescence intensities were collected by constant wavelength single point analysis using SPEX Fluoromax 3 (Horiba Scientific, Piscataway, NJ) with Origin version 8.1.

### Fluorescence polarization kinetic assay

Kinetic polarization assays were performed using alexa[488]-containing secretin probes[22] with WT and mutant SecR membranes, following methods previously reported for cholecystokinin receptor[32]. Fluorescence polarization signals were collected using a Pherastar FSX, following the fluorescence anisotropy protocol (Ex 481 nm, Em 521), with measurements read for 0.5 s/cycle for a total of 200 flashes. Ligand binding was initiated by mixing receptor-enriched membranes (4 μg/mg protein) with alexa[488]-secretin probes (3.2 nM or 10 nM) in a final volume of 200 μl KRH medium pH 7.4 with 0.2% bovine serum albumin for a total of 75 cycles. After 50 cycles, when the polarization signal had reached a plateau, the dissociation of secretin probes was measured following addition of 1 μM unlabeled secretin to prevent rebinding of the fluorescent probe, and continued collection of the signal for another 25 cycles. Specific fluorescence signal was calculated by subtracting non-specific signal (signal in the presence of unlabeled secretin) from the total signal (signal in the absence of competing ligand). The final kinetic data were calculated using non-linear regression curve fitting with equations for association = $Eq*(1-\exp(-1*Kob*X))$ and for dissociation = $Y$ at Time 0* $\exp(-1*Koff*(X - \text{Time } 0))$ where $X =$ time, and $Y =$ total binding using Prism 9.2.

## Fluorescence collisional quenching experiments

Fluorescence collisional quenching studies were performed using the hydrophilic quenching reagent, potassium iodide (KI), as described previously[33]. Fluorescence intensities of receptor-bound probes were collected by exciting samples at 481 nm using Fluoromax-3 spectrofluorometer single point measurement by Origin version 8.1. Emission fluorescence intensity values at 521 nm were collected with an integration time of 10 s, with 4 repetitions for each value, after a sequential addition of KI (1 M aqueous solution in 10 mM $Na_2S_2O_3$). The effect of ionic strength was determined by measuring fluorescence intensities in potassium chloride. Background-subtracted corrected fluorescence data were calculated and plotted based on the Stern–Volmer equation, $F_o/F = 1 + Ksv[Q]$, where $F_o/F$ is the fluorescence intensity in the presence or absence of KI. The Stern–Volmer quenching constant, $K_{SV}$, was calculated from the slope of $F_o/F$ as a function of the quencher concentration $[I^-]$.

## Fluorescence anisotropy studies

Fluorescence anisotropy was measured using an automatic polarizer-equipped L-format-based single-channel Fluoromax-3 spectrofluorometer[22]. Emission measurements were carried out by adjusting the excitation side polarizer to a vertical position (V) with the emission side polarizer in horizontal (H) and vertical (V) positions. Emission and excitation polarizers were aligned for 55° and 0°, respectively. Anisotropy was calculated according to the equation, $A = (I_{VV} - GI_{VH})/(I_{VV} + 2GI_{VH})$, where $I_{VV}$ is the intensity measured with both the excitation- and emission-side polarizers in the vertical positions, and $I_{VH}$ is the intensity measured with the excitation-side polarizer in the vertical position and the emission-side polarizer in the horizontal position. The value of $G$ was calculated by the equation, $G = I_{HH}/I_{HV}$. Measurements were collected by exciting the samples at 481 nm in a temperature-controlled cuvet using the Constant Wavelength Analysis with 10 s integration time and 4 repetitions. Data were collected at 20 and 37 °C in the absence and presence of GppNHp (10 μM), and data were plotted using Prism 9.2.

## Statistical analysis

Comparisons between the experimental and control groups were evaluated using the Mann–Whitney test and one-way ANOVA followed by Dunnett's or Tukey post-test.

## Reporting summary

Further information on research design is available in the Nature Portfolio Reporting Summary linked to this article.

# Data availability

The authors declare that all data supporting the findings of this study are available within the paper and its Supplementary information files. Source data are provided with this paper.

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

## Acknowledgements

This work was supported by a grant from the National Institute of Health, National Institute of General Medical Science [Grant R01 GM132095] (L.J.M.). The authors acknowledge Dr. Xin Zhang for providing the 3DVA data of GLP-1R:sema used in the preparation of Fig. 6. P.M.S. is a Senior Principal Research Fellow of the National Health and Medical Research Council of Australia [NHMRC; ID:1154434]. D.W. is a Senior Research Fellow of the National Health and Medical Research Council of Australia [NHMRC; ID:1155302]. P.M.S., A.C., and D.W. are shareholders in Septerna Inc. and DACRA Tx. The authors would like to acknowledge the technical assistance of J. Milburn, S. Rawal, C. Chen, and M. L. Augustine for this study.

## Author contributions

K.G.H.—performed experiments, data analysis, figure preparation, manuscript preparation. S.J.P.—molecular modeling, figure preparation. A.C.—manuscript preparation. D.W., P.M.S and L.J.M.—project conceptualization, supervision, and manuscript preparation.

## Competing interests

The authors declare no competing interests.
