## [Peer Review File · Nature Communications]

Impact of secretin receptor homo-dimerization on natural ligand bindingReviewer #1 (Remarks to the Author):

The manuscript by Harikumar et al presents a study investigating the dimeric nature of the secretin receptor (SecR) and its implications on the functional potency of the secretin peptide. The authors employ a range of experimental techniques on intact cells and cell membranes, as well as receptor mutants, to elucidate the differences in spatial approximation, binding, and mobility between the wild type SecR and a non-dimerizing mutant. The discussion section provides a comprehensive analysis of the results, suggesting a model where the ability of SecR to form dimers alters the dynamics of the receptor, in a G protein-dependent manner. This study provides some new insights into secretin receptor dimerization and will be of particular interest to the field. Several major concerns from this reviewer need to be addressed.

The most significant concern is the lack of solved dimeric structures for any family member of class B GPCRs. While the authors acknowledge this limitation, but without a homodimeric structure, it is difficult to just extrapolate from biochemical approaches. For example, the study relies heavily on data from cysteine trapping, to probe the spatial approximation and interactions between the secretin peptide and the SecR. What are the potential confounding factors and limitations in this approach. Authors also suggest that the observed differences in peptide mobility between the wild type SecR and the non-dimerizing mutant indicate higher conformational dynamics of the peptide-bound receptor extracellular domain (ECD) and extracellular loops (ECLs) in the dimeric state. However, the link between peptide mobility and G protein recruitment and activation remains speculative. More supporting evidence for this claim are necessary. Another major concern is that the whole study was done in transfected cells overexpressed with SecR and/or mutant, what is the function of receptor dimer in vivo or if it does exist? There were some papers showing potential biologic activity of SecR oligomers in the literature, while there are no further studies subsequently to show a real function of this dimer.

The authors should provide fluorescence images demonstrating successful construction of the fluorescence tool, not just schematics. This will provide additional confidence in the quality of the experimental probes used in the study.

Reviewer #2 (Remarks to the Author):

In this paper the authors report on an extensive investigation of the secretin receptor (SecR), which is a class-B GPCR for which the evidence for dimerization has been quite consistent and convincing. This sets it up as a potentially powerful system for resolving the structural details of dimerization and how dimerization affects receptor activation and downstream signaling. These questions are incredibly important right now as GPCR homo- and hetero-dimerization is widely investigated and is being targeted in many pharmaceutical development efforts. The data in this paper have the potential to resolve a new mechanism for how dimerization effects ligand binding and G protein activation. For these reasons, I think the impact of this paper could be quite high and stimulate many subsequent studies of other GPCR heteromers.

The paper takes advantage of a SecR dimerization deficient mutant (G264A, I268A) for comparison, and measures things like peptide agonist binding and cAMP response to resolve functional differences. They also investigate Cys trapping to assess the conformational dynamics of specific residues within the receptor. Agonist peptide-bound fluorophores are also used to investigate conformational dynamics as well as molecular modelling. They propose a model at the end of the paper in which the secretin peptide interacts differently with WT SecR than with the SecR mutant that does not dimerize. At the core of the model is the idea that dimerized receptors have higher conformational dynamics in the ligand bound state, which in turn leads to increased G protein activation and release. It is very intriguing model, but on that does not seem intuitive to me. I would expect dimerization to reduce the conformational flexibility of the receptors, and so one major revision I would like to see to the manuscript is a chemically detailed proposal for why the dimer is more flexible. Has this been seen previously? Did they also find it surprising?

My second major suggestion is that the authors do some re-writing of the abstract and introduction. I found many of the sentences to be opaque and it wasn't until I read the paper and went back to the abstract and intro that I was able to infer what they were trying to say. For

example, this sentence in the abstract left me totally lost: "Cysteine-trapping demonstrated a shift in spatial approximation of the secretin N-terminal region from predominantly ECL3 at wild type SecR to a pose favoring additional interactions with ECL2 for the non-dimerizing mutant." What is a shift in spatial approximation? These issues arise multiple times in the introduction, and could also be improved in the results sections. I think if the authors could more clearly and simply explain each observation and how it contributes to the model, the paper would be more accessible to the general audience of Nature Comm. If the clarity issues are properly addressed, I think this work has the potential to have a strong impact on the field of GPCR signaling.

One minor comment is that the citations seem inconsistent. Most are superscript, but reference [15] shows up in brackets several times in the results section.

Reviewer #3 (Remarks to the Author):

This study focuses on the secretin receptor (SecR), a prototype class B G protein-coupled receptor (GPCR), and explores its dimeric and monomeric states through a series of well-designed pharmacological, biochemical, and biophysical experiments. The authors present a commendable and in-depth analysis in elucidating the nature of natural secretin peptide interactions in the homodimeric and monomeric complexes of SecR. Of particular note is the use of cysteine trapping to reveal approximate spatial shifts in the secretin N-terminal region. This innovative approach provides valuable insights into the structural dynamics of SecR and its interactions with ligands. Exploration of the C-terminal region of secretin peptides using fluorescent probes is another highlight of this study. The findings regarding the different microenvironment of the intermediate region of secretin in mutant SecR compared to wild type are intriguing and contribute significantly to our understanding of GPCR function. Furthermore, the paper skillfully addresses the complex topic of G-protein interactions and their impact on peptide mobility and receptor dynamics. The discussion of peptide interaction models, supported by empirical data, adds a valuable dimension to our understanding of GPCR dimerization and its functional implications. In terms of methodology, the use of intact cells and cell membranes with receptors is appropriately chosen and well justified, providing a realistic context for the study. The data are robust and well presented, and the conclusions are well supported by the experimental results. Overall, this paper provides new insights into the structure and function of class B G protein-coupled receptors and is a significant contribution to the field. The findings have important implications for drug design and therapeutic interventions targeting these receptors. The study is methodologically sound and the results are clearly expressed and well supported. This paper is worthy of publication in nature communications. However, it needs to be made more accessible to readers who are not GPCR specialists. Specific points raised are as follows.

In your cysteine trapping experiment, you are using the cysteine substituent of Secretin. For readers who are not familiar with SECR, it might be a good idea to know where Cys is introduced in the monomeric experimental structure and where in SecR it is labeled.

You should cite this paper which came out at near the same time as your SECR structure paper (PMID: 33008599).

Although this study is about dimerization of SecR, a comprehensive discussion of the dimerization potential of other class B GPCRs should be included in the discussion. In particular, you should discuss the conservation around TM4, where monomerization mutant (G264A, I268A) have been introduced. It should also be mentioned that RAMP2 has been reported to modulate the activity of glucagon receptor (PMID: 37001505), It is worth comparing the homodimer and heterodimer in terms of modulation of the activity, including their interactions within the membrane.

Reviewer #4 (Remarks to the Author):

The manuscript investigates the nature of Secretin receptor (SecR) dimerization and its role on

receptor activity. Authors use a SecR mutant bearing two mutations in TM4, which previously has been shown to exhibit reduced dimerization propensity in life cell assays compared to WT SecR. Here, with a variety of pharmacological, biochemical and biophysical assays, interaction patterns of different residues of secretin peptides are probed with both receptor variants demonstrating that the mid region of the ligand exhibits higher conformational dynamics in WT SecR vs. mutant SecR. From this, they propose a model where receptor dimerization allows higher local ligand dynamics within the transmembrane domain leading to repetitive receptor activation cycles and thus to increased receptor activity when compared to monomeric receptor.

The study is overall well presented and potentially poses some new and important aspects of class B GPCR function. A current limitation is the use of a receptor mutant as the only proxy for receptor dimerization. Moreover, it remains unclear whether this mechanism may be a class effect or is restricted to SecR.

Specific comments:

- The SecR(G264A, I268A) mutant has previously been shown to exhibit reduced dimerization propensity in life cell assays. However, no measure for dimerization of the receptor variants has been shown in the present study, which may be different under the given experimental conditions. Please provide evidence for receptor dimerization in the current experimental settings.
- Have authors considered, that the TM4 mutations exhibit effects on receptor dynamics independent from receptor dimerization, which may contribute to the observed pharmacological differences?
- At current, it is not clear how the second protomer would affect ligand dynamics in the dimer. Testing ligand dynamics in monomeric vs dimeric receptor state e.g. by MD simulation could further strengthen the hypothesis.
- Fig. 1: Discrimination of individual binding curves is difficult. Colors would be advantageous. Moreover, the characterization of the Alexa-coupled peptides at this stage of the manuscript is somewhat confusing. Authors should consider rearranging the figure.
- Fig. 3: Positions for peptide crosslinking between both receptor variants seem to be mutually exclusive. Were none of the positions crosslinked in both receptor variants?
- Authors compare Cys-trapping experiments on mutant SecR from the present study with those on wt SecR from a previous study. Were the experimental conditions comparable? This needs to be stated clearly in the manuscript.

Minor comments:

- The Reporting Summary apparently does not belong to the current manuscript.
- The description of membrane preparation in the methods section is too brief. Please provide more details on which cell lines have been used and on the procedure itself.
- Methods receptor binding assays: "After the incubation, cells were washed twice with ice-cold KRH medium to separate bound from free radioligand before being lysed with 0.5M NaOH or membrane-bound and free ligand were separated by centrifugation." How were cells washed? By centrifugation or filtration? Were membrane fractions not washed?
- Page 13: "the N-terminal alexa was more easily quenched than the alexa in other position within the peptide" should be rephrased.
- Add a symbol to Fig 4 to indicate orientation of the two views.

Re: NCOMMS-23-55365

Responses to reviewers:

Reviewer #1 (Remarks to the Author)

The manuscript by Harikumar et al presents a study investigating the dimeric nature of the secretin receptor (SecR) and its implications on the functional potency of the secretin peptide. The authors employ a range of experimental techniques on intact cells and cell membranes, as well as receptor mutants, to elucidate the differences in spatial approximation, binding, and mobility between the wild type SecR and a non-dimerizing mutant. The discussion section provides a comprehensive analysis of the results, suggesting a model where the ability of SecR to form dimers alters the dynamics of the receptor, in a G protein-dependent manner. This study provides some new insights into secretin receptor dimerization and will be of particular interest to the field. Several major concerns from this reviewer need to be addressed.

Thank you for your comments and recognition that this provides new insights that will be of particular interest to the field.

The most significant concern is the lack of solved dimeric structures for any family member of class B GPCRs. While the authors acknowledge this limitation, but without a homodimeric structure, it is difficult to just extrapolate from biochemical approaches. For example, the study relies heavily on data from cysteine trapping, to probe the spatial approximation and interactions between the secretin peptide and the SecR. What are the potential confounding factors and limitations in this approach. Authors also suggest that the observed differences in peptide mobility between the wild type SecR and the non-dimerizing mutant indicate higher conformational dynamics of the peptide-bound receptor extracellular domain (ECD) and extracellular loops (ECLs) in the dimeric state. However, the link between peptide mobility and G protein recruitment and activation remains speculative. More supporting evidence for this claim are necessary. Another major concern is that the whole study was done in transfected cells overexpressed with SecR and/or mutant, what is the function of receptor dimer in vivo or if it does exist? There were some papers showing potential biologic activity of SecR oligomers in the literature, while there are no further studies subsequently to show a real function of this dimer.

We agree that the link between peptide mobility and G protein recruitment and activation remains somewhat speculative, but it is entirely consistent with what we understand structurally about lower affinity but high efficacy peptide agonists at the GLP-1R (as noted in the manuscript). We believe that the speculation is appropriate in the context of the extant data presented in our manuscript and correlative data from the literature.

As we note in the Introduction and Discussion, no one has yet reported high resolution structures for any dimeric complexes involving class B GPCRs. Indeed, the data presented in the current report provides a potential explanation for the difficulties in solving such structures, particularly as active-state complexes. We have modified the paper to better explain how each of the component series of techniques applied in this work contributes to this understanding. The fluorescence data complement the crosslinking data and are internally consistent. We have been working towards determination of a structure of the SecR dimeric complex and have low resolution cryo-EM data of the dimer demonstrating that this can be extracted and purified, but this is not yet ready for publication and is beyond the scope of the current report.

We also now add background information to address the concern about receptor overexpression in the demonstration of class B GPCR dimerization. In our model cell studies, we have used not only receptor-overexpressing cell lines, but also used inducible systems that varied the density of receptors on the cells starting with subphysiologic levels of expression (Ward et al. *Biochem J* 474:1879, 2017; Asher et al. *Nature Meth* 18:397, 2021). Additionally, we have systematically explored the effect of receptor densities of SecR and GLP-1R in model systems on formation of hetero-dimeric receptor complexes (Harikumar et al. *Endocrinology* 158:1685, 2017). We have also demonstrated hetero-dimerization between these two class B GPCRs in naturally-occurring receptors on pancreatic islets, where we demonstrated a functional effect (Harikumar et al. *Endocrinology* 158:1685, 2017). This work is discussed in the revised manuscript.

As noted, TM4-mediated dimerization of class B GPCRs appears to be a class effect. To gain further evidence for the physiological importance of class B GPCR dimerization, using the related GLP-1R, we generated a mouse model in which the WT GLP-1R is replaced by its TM4 mutant that disrupts its dimerization. While outside of the scope of the current manuscript, this mouse exhibits a prominent phenotype in the context of diet-induced obesity. Collectively, we are confident of the critical importance of dimerization for the SecR and more broadly for class B GPCRs.

The authors should provide fluorescence images demonstrating successful construction of the fluorescence tool, not just schematics. This will provide additional confidence in the quality of the experimental probes used in the study.

Requested images of the four fluorescence probes bound to SecR in the absence and presence of competing secretin have been added as Supplementary Figure 1 to the manuscript.

Reviewer #2 (Remarks to the Author)

In this paper the authors report on an extensive investigation of the secretin receptor (SecR), which is a class-B GPCR for which the evidence for dimerization has been quite consistent and convincing. This sets it up as a potentially powerful system for resolving the structural details of dimerization and how dimerization affects receptor activation and downstream signaling. These questions are incredibly important right now as GPCR homo- and hetero-dimerization is widely investigated and is being targeted in many pharmaceutical development efforts. The data in this paper have the potential to resolve a new mechanism for how dimerization effects ligand binding and G protein activation. For these reasons, I think the impact of this paper could be quite high and stimulate many subsequent studies of other GPCR heteromers.

Thank you for your kind comments.

The paper takes advantage of a SecR dimerization deficient mutant (G264A, I268A) for comparison, and measures things like peptide agonist binding and cAMP response to resolve functional differences. They also investigate Cys trapping to assess the conformational dynamics of specific residues within the receptor. Agonist peptide-bound fluorophores are also used to investigate conformational dynamics as well as molecular modelling. They propose a model at the end of the paper in which the secretin peptide interacts differently with WT SecR than with the SecR mutant that does not dimerize. At the core of the model is the idea that dimerized receptors have higher conformational dynamics in the ligand bound state, which in turn leads to increased G protein activation and release. It is very intriguing model, but on that does not seem intuitive to me. I would expect dimerization to reduce the conformational flexibility

of the receptors, and so one major revision I would like to see to the manuscript is a chemically detailed proposal for why the dimer is more flexible. Has this been seen previously? Did they also find it surprising?

Given the conformational flexibility/plasticity of GPCRs, there is no good way to predict the dynamics. It is likely that some parts of the receptor are less dynamic in the dimer, however, the ECD is often observed to be highly dynamic in active structures of the related GLP-1R with peptides that have more transient interactions with the deep binding pocket. As our observations were restricted to the regions engaged by the peptide probes, we cannot comment on other parts of the receptor. We had previously observed the higher affinity and potency of agonists at the SecR homo-dimer, but did not know how to explain it at that time. The dynamics informed by the fluorescence studies seem to provide a good explanation for this that we have developed in the current Discussion. It is well established that there is allosteric communication between the receptor and G protein, so it is not surprising that there are differential effects when the receptor is decoupled. Moreover, in studies with the class B calcitonin GPCR, we have previously observed that formation of the ternary complex with the G protein is associated with destabilization of the dimer (Furness et al, Cell 2016), which is consistent with our model and in difficulties in obtaining active-state, G protein-coupled receptor dimer structures. However, we are unaware of similar studies directly probing receptor dynamics.

My second major suggestion is that the authors do some re-writing of the abstract and introduction. I found many of the sentences to be opaque and it wasn't until I read the paper and went back to the abstract and intro that I was able to infer what they were trying to say. For example, this sentence in the abstract left me totally lost: "Cysteine-trapping demonstrated a shift in spatial approximation of the secretin N-terminal region from predominantly ECL3 at wild type SecR to a pose favoring additional interactions with ECL2 for the non-dimerizing mutant." What is a shift in spatial approximation? These issues arise multiple times in the introduction, and could also be improved in the results sections. I think if the authors could more clearly and simply explain each observation and how it contributes to the model, the paper would be more accessible to the general audience of Nature Comm. If the clarity issues are properly addressed, I think this work has the potential to have a strong impact on the field of GPCR signaling.

We apologize for the opacity of some of our previous descriptions of methodology. We have modified this to help the general audience better understand the methods utilized.

One minor comment is that the citations seem inconsistent. Most are superscript, but reference [15] shows up in brackets several times in the results section.

We have corrected the inconsistency in reference format noted.

Reviewer #3 (Remarks to the Author)

This study focuses on the secretin receptor (SecR), a prototype class B G protein-coupled receptor (GPCR), and explores its dimeric and monomeric states through a series of well-designed pharmacological, biochemical, and biophysical experiments. The authors present a commendable and in-depth analysis in elucidating the nature of natural secretin peptide interactions in the homodimeric and monomeric complexes of SecR. Of particular note is the use of cysteine trapping to reveal approximate spatial shifts in the secretin N-terminal region. This innovative approach provides valuable insights into the structural dynamics of SecR and its interactions with ligands. Exploration of the C-terminal region of

secretin peptides using fluorescent probes is another highlight of this study. The findings regarding the different microenvironment of the intermediate region of secretin in mutant SecR compared to wild type are intriguing and contribute significantly to our understanding of GPCR function. Furthermore, the paper skillfully addresses the complex topic of G-protein interactions and their impact on peptide mobility and receptor dynamics. The discussion of peptide interaction models, supported by empirical data, adds a valuable dimension to our understanding of GPCR dimerization and its functional implications. In terms of methodology, the use of intact cells and cell membranes with receptors is appropriately chosen and well justified, providing a realistic context for the study. The data are robust and well presented, and the conclusions are well supported by the experimental results. Overall, this paper provides new insights into the structure and function of class B G protein-coupled receptors and is a significant contribution to the field. The findings have important implications for drug design and therapeutic interventions targeting these receptors. The study is methodologically sound and the results are clearly expressed and well supported. This paper is worthy of publication in nature communications. However, it needs to be made more accessible to readers who are not GPCR specialists. Specific points raised are as follows.

We really appreciate your kind comments and suggestions. We have revised the descriptions of the methodology to make it more accessible to a general audience.

In your cysteine trapping experiment, you are using the cysteine substituent of Secretin.

The positions of cysteine incorporation have been better described in the revision.

For readers who are not familiar with SECR, it might be a good idea to know where Cys is introduced in the monomeric experimental structure and where in SecR it is labeled.

This, too, has been better communicated.

You should cite this paper which came out at near the same time as your SECR structure paper (PMID: 33008599).

Reference added.

Although this study is about dimerization of SecR, a comprehensive discussion of the dimerization potential of other class B GPCRs should be included in the discussion. In particular, you should discuss the conservation around TM4, where monomerization mutant (G264A, I268A) have been introduced. It should also be mentioned that RAMP2 has been reported to modulate the activity of glucagon receptor (PMID: 37001505), It is worth comparing the homodimer and heterodimer in terms of modulation of the activity, including their interactions within the membrane.

Discussion has been expanded to include this reference and to expand what we now understand about dimerization across this family of GPCRs.

Reviewer #4 (Remarks to the Author):

The manuscript investigates the nature of Secretin receptor (SecR) dimerization and its role on receptor activity. Authors use a SecR mutant bearing two mutations in TM4, which previously has been shown to exhibit reduced dimerization propensity in life cell assays compared to WT SecR. Here, with a variety of pharmacological, biochemical and biophysical assays, interaction patterns of different residues of

secretin peptides are probed with both receptor variants demonstrating that the mid region of the ligand exhibits higher conformational dynamics in WT SecR vs. mutant SecR. From this, they propose a model where receptor dimerization allows higher local ligand dynamics within the transmembrane domain leading to repetitive receptor activation cycles and thus to increased receptor activity when compared to monomeric receptor.

The study is overall well presented and potentially poses some new and important aspects of class B GPCR function. A current limitation is the use of a receptor mutant as the only proxy for receptor dimerization. Moreover, it remains unclear whether this mechanism may be a class effect or is restricted to SecR.

Thank you very much for your kind comments. We include evidence that TM4-mediated dimerization is a class effect that appears to be relevant to every member of this family studied to date. As noted above, we have prepared a mouse model for disruption of dimerization of the GLP-1R and have observed a notable phenotype that we hope to report in the near future.

Specific comments:

- The SecR(G264A, I268A) mutant has previously been shown to exhibit reduced dimerization propensity in life cell assays. However, no measure for dimerization of the receptor variants has been shown in the present study, which may be different under the given experimental conditions. Please provide evidence for receptor dimerization in the current experimental settings.

The experimental conditions utilized in the current work are identical to those in which the SecR(G264A,I268A) construct has previously been fully characterized. Additionally, this TM4 mutant of SecR has also been utilized in then laboratories of Jonathan Javitch (Asher et al. Nature Meth 18:397, 2021), Graeme Milligan (ward et al. Biochem J 474:1879, 2017), and Vali Raicu (Biener et al. Biophys J 120:3028, 2021), who utilized a variety of experimental approaches that all reflected similar disruption of dimerization of this construct.

- Have authors considered, that the TM4 mutations exhibit effects on receptor dynamics independent from receptor dimerization, which may contribute to the observed pharmacological differences?

The process of dimerization is dynamic and in equilibrium with the monomeric state of SecR. Our data are indeed compatible with shifting the equilibrium to make the dimeric state less likely. All functional studies to date have shown that this construct is still capable of ligand binding, biological signaling, and cell trafficking.

- At current, it is not clear how the second protomer would affect ligand dynamics in the dimer. Testing ligand dynamics in monomeric vs dimeric receptor state e.g. by MD simulation could further strengthen the hypothesis.

Meaningful MD analysis would require a credible molecular model of the dimeric complex of SecR. We believe this is well beyond the scope of the current work.

- Fig. 1: Discrimination of individual binding curves is difficult. Colors would be advantageous. Moreover, the characterization of the Alexa-coupled peptides at this stage of the manuscript is somewhat confusing. Authors should consider rearranging the figure.

The figure has been revised to include colors to better delineate the individual curves.

- Fig. 3: Positions for peptide crosslinking between both receptor variants seem to be mutually exclusive. Were none of the positions crosslinked in both receptor variants?

This figure highlights only those residues with the highest level of covalent labeling. Table 2 shows the percentage labeling of each residue within ECL2 and ECL3, illustrating that the residues previously labeled for WT SecR are indeed labeled in the TM4 mutant as well, but were not the dominant sites labeled. This is consistent with the model of altered dynamics that we have proposed.

- Authors compare Cys-trapping experiments on mutant SecR from the present study with those on wt SecR from a previous study. Were the experimental conditions comparable? This needs to be stated clearly in the manuscript.

Conditions were identical. This has been clarified.

Minor comments:

- The Reporting Summary apparently does not belong to the current manuscript.

Reporting Summary updated.

- The description of membrane preparation in the methods section is too brief. Please provide more details on which cell lines have been used and on the procedure itself.

Methods have been expanded.

- Methods receptor binding assays: "After the incubation, cells were washed twice with ice-cold KRH medium to separate bound from free radioligand before being lysed with 0.5M NaOH or membrane-bound and free ligand were separated by centrifugation." How were cells washed? By centrifugation or filtration? Were membrane fractions not washed?

Methods have been clarified.

- Page 13: "the N-terminal alexa was more easily quenched than the alexa in other position within the peptide" should be rephrased.

Sentence revised.

- Add a symbol to Fig 4 to indicate orientation of the two views.

Figure 4 orientation better described.

Reviewer #2 (Remarks to the Author):

The revisions to this manuscript have significantly improved its readability. The significance of the dynamic states captured in this extensive study is easier to understand and I am satisfied with the new version of the manuscript. I support its publication.

Reviewer #3 (Remarks to the Author):

All my requirements have been met and I recommend it for publication in nature communications.

Reviewer #4 (Remarks to the Author):

The authors have adequately addressed my concerns.